# Gender Impact on Diabetes Distress Focus at Medical Communication Concerns, Life and Interpersonal Stress

**DOI:** 10.3390/ijerph192315678

**Published:** 2022-11-25

**Authors:** Li-Chi Huang, Ching-Ling Lin, Yao-Tsung Chang, Ruey-Yu Chen, Chyi-Huey Bai

**Affiliations:** 1Endocrinology & Metabolism, Cathay General Hospital, Taipei 106438, Taiwan; 2School of Public Health, Taipei Medical University, Taipei 110301, Taiwan; 3School of Medicine, College of Medicine, Taipei Medical University, Taipei 110301, Taiwan; 4School of Medicine, National Tsing Hua University, Hsinchu 300044, Taiwan

**Keywords:** diabetes distress, culture, gender difference, scale

## Abstract

Introduction: along with the rapidly aging population in many countries around the world, the global prevalence of diabetes and suffering from diabetes-related depression have risen in middle-aged and elderly adults. However, given that psychological stress is deeply influenced by culture, gender inequality in these statistics is often exhibited and increases with age. The aim of this study was to explore the gender difference in diabetes distress among middle-aged and elderly diabetic patients. Methods: 395 participants from four hospitals were recruited for a cross-sectional survey. The Taiwan Diabetes Distress Scale (TDDS) was used to measure diabetes distress. Linear regression was conducted to assess the gender difference in different types of diabetes distress. Results: there was significant gender difference in each diabetes distress domain. In particular, men who had received diabetes education in the past six months seemed to be more concerned about diabetes complications and felt pressured to communicate with doctors. In addition, women seemed to be more affected by diabetes distress because of their marital status, especially for married women. Conclusions: diabetes distress seems to have significant gender differences; however, more longitudinal research is needed on the causal relationship between gender and diabetes distress.

## 1. Introduction

Diabetes is a chronic disease which can result in comorbidities and mortality, imposing a heavy burden on medical and social economic systems worldwide. As we are facing a rapidly aging population in many countries around the world, the World Health Organization (WHO) states that the global prevalence of diabetes in adults has risen to 8.5% [1]. At the same time, the WHO also states that people with diabetes have a two–three times higher risk of suffering from depression than those without diabetes [2]. However, “diabetes distress” may be an issue that calls for more attention in the diagnosis of depression.

Diabetes distress is defined as emotional burdens, stress, and worries associated with the demanding chronic disease, blood glucose control, or complications of diabetes [3]. These psychological burdens and worries may further disturb patients’ mental health and behavior, but not to the extent of depression or anxiety. In the past few years, some studies have explored this issue; however, given that psychological stress and depression are deeply affected by race and culture, it is necessary to specifically discuss the issue of diabetes distress in Chinese people. For example, many studies have pointed out that the traditional values in Chinese culture and norms of Chinese immigrants are different from those of Westerners, which can play a role in affecting psychological stress and even depression, and should be taken into account when performing psychotherapy [4,5]. Ethnic and cultural differences also directly affect diabetes self-management behaviors and treatment methods [6]; therefore, for diabetes distress and diabetes care, it is necessary to conduct in-depth discussions based on the cultural characteristics of the Chinese, and the results obtained from studies based on other ethnicity groups may not be directly applicable.

Based on the above consideration, the two current mainstream diabetes distress questionnaires, Problem Areas in Diabetes (PAID) and Diabetes Distress Scale (DDS), may be less suitable for Chinese culture (even if both have been translated into Chinese). We have developed a new tool suitable for assessing the Taiwanese [7]. The Taiwan Diabetes Distress Scale (TDDS) has suitable reliability and validity, and, in particular, it can reflect the unique doctor–patient relationship and communication concerns of the Chinese. Some studies found that Chinese people tend to over-report their medication adherence, so as not to disappoint their healthcare providers [8]. Since we have indeed found that many patients are afraid to tell the truth to their physicians, or care more about what other people think of them than how good their diabetes is controlled, these factors were taken into consideration while we developed the TDDS.

The previous literature on psychological stress has suggested that such characteristics may also have gender differences. Gender inequality is often exhibited and increases with age. The purpose of this study was to perform a secondary analysis based on the psychological and behavioral indicators which we collected during the TDDS questionnaire development to explore gender differences in diabetes distress in middle-aged and elderly diabetic patients in Taiwan.

## 2. Materials and Methods

Based on the data collected during the development of the TDDS, which has been published elsewhere [7], we conducted a secondary analysis that focused on gender impacts. Data were collected from division hospitals with the Cathay Medical system in Northern Taiwan between September and November 2021. The study protocol was approved by the Institutional Review Board (IRB) of Cathay General Hospital.

### 2.1. Study Population and Features

Considering the difference in regions, types of hospitals, and patients’ characteristics, we applied the cluster randomized sampling method to allocate the sampling proportions according to the number of patients with type 2 diabetes in the four hospitals. There were 395 valid samples for TDDS development. The Cathay Medical system currently has four hospitals in Taiwan, namely a medical center, two regional hospitals, and one clinic, three of them located in Taipei and the fourth one in Hsinchu. In accordance with the proportion of the total number of patients with type 2 diabetes in each hospital, 170, 100, 80, and 50 questionnaires were retrieved from Taipei, Sijhih Branch, Hsinchu Branch, and Neihu Clinic of the Cathay Medical system, respectively.

### 2.2. Measures

Diabetes distress was measured by the TDDS, which contained 12 items in three domains, namely: fear of diabetes complications, communication concerns, and life and interpersonal stress. We developed this Traditional Chinese questionnaire in 2021. The TDDS has good construct validity and can explain about 75% of the variation, and the Cronbach’s α of each domain is between 0.863 and 0.924. Every item began with the heading “I am worried about…”, “Because of…, it makes me feel stressed”, or “When I…, it makes me feel anxious”. Items were rated on a scale of 0–10 points; 0 denoted the least worried or least stressed while 10 denoted extremely worried or extremely stressed. The items and total scores of the three subscales were: fear of diabetes complications: 3 items, 0–30 points; communication concerns: 3 items, 0–30 points; life and interpersonal stress: 6 items, 0–60 points.

Healthy lifestyle variables were comprised of physical activity, vegetable consumption, fruit consumption, regular diet diary, and regular self-monitoring blood glucose (SMBG). Physical activity was measured with the Godin leisure-time physical activity scale [9,10], which marks the number of days in a week a subject does vigorous (9 points), medium (5 points), and light (3 points) physical activities. After weighing and summing up each level of physical activity, it defines an active person as one with a total of ≥24 points, a moderately active person as one with 14~23 points, and an insufficiently active person as one with ≤13 points. Both vegetable and fruit consumption levels were measured by a single item, which evaluated the number of servings per day consumed in the past week. Both regular diet diary and regular SMBG were measured by a single item, which evaluated the number of days of the week these behaviors occurred.

Health status variables included the duration of diabetes treatment (years), blood glucose control (HbA1c, %), whether diabetes education and nutrition education had been implemented within the past six months, and other chronic diseases or diabetes complications (5 items, including hyperlipidemia, hypertension, arthritis, kidney disease, and others). Sociodemographic variables included gender, age, educational level, and marital status.

### 2.3. Statistical Analysis

A *t*-test and Chi-squared test were employed to assess differences in sociodemographic factors, healthy lifestyles, health statuses, and diabetes distress among males and females at the baseline (Table 1). Before multiple variable analysis, we conducted Pearson correlation analysis to screen variables suitable for regression analysis. Since most health behavior and health status indicators were not significantly correlated with diabetes distress, only a few variables were finally included in the analysis (Table 2 and Table 3). Then, we conducted linear regression to analyze the risk factors of medium or high diabetes distress and possible gender differences.

## 3. Results

Table 1 shows the demographic characteristics of the 395 patients. In total, 54.7% were males, the mean age was 57.53 years (SD = 12.68), 48.2% had a bachelor’s degree or above, and male participants had a higher education level than females. Their mean HbA1c was 7.19% (SD = 1.30), their duration of diabetes was 10.22 years (SD = 8.23), 37.0% had hypertension, 31.1% had hyperlipidemia, and 6.6% had chronic kidney disease, all without gender differences in prevalence. The mean diabetes distress was 47.42 points (SD = 28.21), and 32.4% had a tendency to have diabetes distress. Women had significantly higher diabetes distress than men, and the total score and scores from each domain were all significantly higher than men. In terms of health behaviors, there were no gender differences in fruit and vegetable consumption, frequency of SMBG, and diet diaries. Of the participants, 45.7% never monitored their blood glucose, and 79.2% never kept a record of their diet. There was no gender difference in health education and examinations related to diabetes. Overall, 64.3% of the participants had received education from a diabetic educator in the past six months, and 35.2% of them had been interviewed by a dietitian. These variables are not listed in the Table 1 as there are no gender differences.

The results of the regression analysis of diabetes distress are listed in Table 2. Patients who were female (*p* < 0.001), younger (*p* < 0.001), with longer duration of diabetes (*p* = 0.027), and with higher HbA1c (*p* < 0.001) had a significantly higher risk of having diabetes distress (Model 1). In terms of gender impact (Model 2 and 3), men who were younger (*p* = 0.021) and who had higher HbA1c (*p* < 0.001) were at higher risk of having diabetes distress. However, in women, being younger (*p* < 0.001), longer duration of diabetes (*p* = 0.011), and being married (*p* = 0.016) were significant predictors of having higher diabetes distress, in addition to HbA1c (*p* < 0.001). The explained variation in the three models was between 10.7% and 15.3%, which might mean that there may be other important diabetes distress factors that have not been discovered or discussed.

Males and females responded differently to different aspects of diabetes distress (Table 3). Men who had higher HbA1c (*p* = 0.037), recent diabetes education (*p* = 0.033), and reluctance in taking SMBG (*p* = 0.050), had a greater fear of diabetes complications; however, for women, only HbA1c (*p* < 0.001) had an impact on such fears. Men who had higher HbA1c (*p* < 0.01), recent nutrition education (*p* = 0.006), and chronic kidney diseases (*p* = 0.047) showed higher diabetes distress related to communication concerns; however, for women, only HbA1c (*p* < 0.001) predicted higher distress from communication concerns. For men, those who were younger (*p* < 0.001) and who had higher HbA1c (*p* < 0.001) had higher diabetes life and interpersonal stress; however, for women, the influence of being married (*p* < 0.001) on this stress was stronger than males.

## 4. Discussion

In this study, we found that there were gender differences in diabetes distress. Obviously, males and females have different perceptions of and feelings about diabetes distress. According to the overall TDDS criteria for distinguishing diabetes distress, blood glucose control (HbA1c) is the main source of distress, although, given that its overall predictive power was not high (<20%), there may be other factors associated with diabetes distress, and these factors might also have gender differences. In recent years, some studies have found that females were more likely to have diabetes distress [11,12], and, in our study, the risk of diabetes distress was 2.67 times higher (odds ratio) in females than males, which is consistent with the literatures.

This difference may be more pronounced in different domains of our TDDS. It seems that, for men, receiving diabetes education but failing to perform SMBG regularly may make them fear diabetes complications more. Studies have found that men seem to be more likely to underestimate diabetes-related problems than women [13]; in addition, studies have also found that women are more proactive in self-management and preventive care, searching for relevant information and adapting to chronic diseases, while men are more accustomed to dealing with urgent problems [14]. Since there are more men who have long-ignored the seriousness of diabetes and its complications, only when they have higher blood glucose and are warned by doctors do they start to feel afraid of their diabetes. Hence, it is inevitable that men experience greater diabetes distress when the problem is so serious that they must face it. Although there are many studies on these gender differences, these differences in behaviors and attitudes have rarely been linked to diabetes distress in the past. It is not clear whether there are cultural differences in these findings; however, in our study, Taiwanese people seem to show gender differences in their fears regarding diabetes complications.

This study found that, apart from the degree of blood glucose control, men who had recently received nutrition education and had been diagnosed with chronic kidney disease might have a greater fear of communicating with their physicians significantly, while women seemed to have other factors that would affect their attitudes to communicate with physicians. Such gender differences may also be the reason why men and women reported experiencing varied pressures to engage in diabetes communication. In the past, many studies have pointed out that there are gender differences in medical communication, and the general communication and comprehension styles of men and women are different [15]. The latest research also pointed out that the words used by medical staff to describe diabetes-related matters have different positive and negative effects on patients’ feelings [16]. For example, women require more empathy from physicians for diabetes communication [17] because they tend to have lower self-efficacy than men in facing diabetes [18]; hence, there must be different communication methods for both genders, and it may even be necessary to use different terms for both genders to describe matters related to diabetes. These factors obviously also affect distressed diabetic patients in the communication between doctors and patients. According to these studies, we may speculate that men are more ashamed to face doctors and dare not communicate with doctors because of their poor blood glucose control, while women are more likely to have physician–patient communication-related diabetes distress if the physician is limited in communication skills and empathy. However, since this study did not delve into these communication competency issues, it can only be left for future research.

In the domain of life and interpersonal stress, this study found that the most obvious difference between the two genders is the influence of marital status. It appears that married women have significantly higher diabetes distress, while men are unaffected by marital status. It is well known that family support provided by a spouse is very important for diabetes self-management [19]. Good marital and family support will strengthen patients’ positive comprehension and attitudes toward chronic diseases [20], and significantly improves self-care behaviors and, thus, improve disease control [21]. For diabetes self-management, studies have also demonstrated that gender is the principal factor in the relationship between spousal support and dietary adherence [22]. Many studies have found that divorced or separated men have a significantly higher risk of diabetes death than married men, which might indicate that married men are in the status of care recipients [23]. For traditional Taiwanese families, because women are mostly the main caregivers, and they often have to juggle work and caring, they usually experience higher family and career conflicts [24]. Therefore, in the domain of life- and interpersonal-related diabetes distress, it is quite plausible that women are more significantly affected by marital status. That is to say, the quality of the marital relationship will significantly affect women facing diabetes and diabetes distress.

Compared with previous studies, this study did not find a relationship between healthy behaviors, such as eating habits and physical activity, and diabetes distress [12,25]; however, what appears to be consistent with the literature is that behaviors that are more directly related to blood sugar management, such as SMBG and medication compliance, are still significantly associated with diabetes distress [11]. Thus far, the evidence on whether focus on health behavior implementation can effectively improve diabetes distress is still inconsistent [26], and the reason may be that diabetes distress is mainly affected by the degree of glycemic control. Therefore, healthy behavior intervention must first effectively improve glycemic control before alleviating diabetes distress. In addition, considering that the quality of doctor–patient communication may also have an impact on the patients’ psychological stress level [16], targeting the psychological needs of different ethnic groups such as empathy and comprehension style, as well as practical solutions for high blood glucose, may be more effective in reducing stress. Given that there are not many longitudinal studies with sufficient quality of evidence related to diabetes distress [27,28], whether there is a bidirectional vicious cycle between poor glycemic control and diabetes distress and the impact of psychodynamics of healthy living toward diabetes distress needs further study.

Therefore, based on the findings of this study, we have several suggestions for future research on diabetes distress in Taiwan. First, since men and women have different attitudes and behaviors toward diabetes distress, whether physicians and diabetic educators should adopt different communication methods based on gender to help reduce diabetes distress needs further exploration. Second, since diabetes distress seems to be influenced by ethnicity and culture, and given that there has been far less research on physician–patient communication and chronic disease management among the Chinese population in Asia than in the West, it is necessary to conduct an in-depth study of diabetes distress among Chinese patients. Third, in view of the possible impact of diabetes distress on glycemic control, longitudinal studies should be conducted in the future, which include more process and outcome indicators to explore the impact of diabetes distress on diabetes treatment.

There are several strengths in this study. First, this study is the first to explore diabetes distress specifically for the Chinese population. Given that Chinese people have unique cultures and customs that may affect their diabetes management, ethnically targeted research may be of considerable value. Second, this study used a tool developed specifically for Taiwanese people; therefore, it may more accurately explore the results than other diabetes distress measurement tools. We believe that it is necessary to develop and use local tools when exploring behavioral and psychological issues that may be influenced by ethnicity and culture.

The limitations of this study could be used to build future research. First, the cross-sectional design did not allow us to draw causal relationships among these variables. Second, although we conducted surveys in four differently-scaled hospitals, the four hospitals were all located in northern Taiwan. Given that there may be differences in patients’ characteristics between urban and rural areas, future surveys should be conducted from broader island-wide samples.

## 5. Conclusions

In this study, we found that there may be gender differences in diabetes distress. Men seem to be more concerned about diabetes complications and feel pressured to communicate with doctors if they received diabetes education in the past six months. Women seem to be more affected by diabetes distress based on their marital relationship. Based on our findings and on the limitations of this study, we suggest that a longitudinal study of diabetes distress in Taiwan be conducted in the future to explore its causal relationships.

## Figures and Tables

**Table 1 ijerph-19-15678-t001:** Demographic characteristics and health status of samples.

	Cathay Hospital Systems		
	Male (*n* = 216), N(%)	Female (*n* = 179), N(%)	*p* Value
Age, years			0.160
18–39	19(8.8)	18(10.1)	
40–49	43(20.0)	21(11.8)	
50–64	85(39.5)	82(46.1)	
65 or higher	68(31.6)	57(32.0)	
Age, mean ± SD	56.84 ± 12.67	58.37 ± 12.68	0.235
Education level			<0.001
Junior high school and bellow	29(13.5)	57(31.8)	
Senior high school	62(28.8)	56(31.3)	
University	99(46.0)	61(34.1)	
Master’s degree or above	25(11.6)	5(2.8)	
Marital status			0.892
Unmarried	33(15.3)	28(15.6)	
Married	172(79.6)	140(78.2)	
Others	11(5.1)	11(6.1)	
Duration of diabetes treatment (years, mean ± SD)	9.95 ± 8.11	10.53 ± 8.37	0.489
HbA1c (%, mean ± SD)	7.07 ± 1.08	7.32 ± 1.51	0.055
Diabetes distress, mean ± SD			
Total point	42.87 ± 25.71	52.91 ± 30.14	<0.001
Fear of diabetes complications	14.48 ± 8.27	17.76 ± 9.46	<0.001
Communication concerns	10.89 ± 7.76	13.22 ± 9.12	0.006
Life and interpersonal stress	17.50 ± 13.59	21.93 ± 16.06	0.003
Have diabetes education	140(64.8)	114(63.7)	0.816
Have nutrition education	77(35.6)	62(34.6)	0.834
Self-monitoring blood glucose			0.888
Never	100(46.5)	80(44.7)	
one or more times per week	115(53.5)	99(55.3)	
Diet record			0.964
Never	168(78.1)	144(80.4)	
one or more times per week	47(21.9)	35(19.6)	
Chronic diseases			
Hypertension	83(38.4)	63(35.2)	0.508
Hyperlipidemia	62(28.7)	61(34.1)	0.251
Kidney disease	16(7.4)	10(5.6)	0.468

**Table 2 ijerph-19-15678-t002:** Regression analysis of diabetes distress (total point).

	Regression Analysis β(SE)		
	Diabetes Distress		
	All sample	Male	Female
	Model 1	Model 2	Model 3
Gender	9.631 ** (2.737)		
Age (years)	−0.494 ** (0.123)	−0.316 * (0.136)	−0.832 ** (0.219)
Duration of diabetes	0.414 * (0.187)		0.745 * (0.290)
Marital status			13.017 * (5.371)
HbA1c	5.968 ** (1.055)	6.241 ** (1.580)	5.957 ** (1.422)
Diabetes health education		7.233 * (3.573)	
R^2^	0.144	0.107	0.153

*: 0.05 > *p* > 0.01, **: 0.01 > *p.*

**Table 3 ijerph-19-15678-t003:** Regression analysis of diabetes distress (subscale).

	Diabetes Distress β(SE)					
	Fear of Diabetes Complications		Communication Concerns		Life and Interpersonal Stress	
	Male	Female	Male	Female	Male	Female
Age					−0.233 ** (0.072)	−0.353 ** (0.102)
Marital status						5.555 * (3.075)
HbA1c	1.099 ** (0.524)	1.615 * (0.463)	2.118 ** (0.471)	1.536 ** (0.446)	3.018 ** (0.833)	3.108 ** (0.759)
Diabetes education	2.546 * (1.184)					
Nutrition education			2.970 ** (1.064)			
SMBG	0.457 * (0.232)					
CKD			3.910 * (1.955)			
R^2^	0.054	0.066	0.142	0.059	0.098	0.128

SMBG = Self-monitoring blood glucose. CKD = Chronic kidney disease. *: 0.05 > *p* > 0.01, **: 0.01 > *p.*

## Data Availability

The data presented in this study are available on request from the corresponding author. The data are not publicly available due to the Institutional Review Board of Cathay General Hospital privacy protection policy.

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
