# Peer review of "Gender Impact on Diabetes Distress Focus at Medical Communication Concerns, Life and Interpersonal Stress"

_ijerph, 2022, doi:10.3390/ijerph192315678_

Round 1

Reviewer 1 Report

Please find the comments in the attachment

Author Response

  1. Is this instrument available to be shared in an appendix?

We can only share the Chinese version since it hasn’t been translated into an English version yet.

  1. Methods: Could you provide samples of your measurement instruments as appendices?

        As above. They are in Chinese only. 

  3. Results: any thought given to using logistic regression?

Since the original design of TDDS is to collect the total score, we first analyzed it with linear regression, as shown in Table 2, and then we used logistic regression as shown in Table 3. We believe that these two different analyses can present different results.

4. Discussion: I am not sure I know what you mean by "varied pressures to engage in diabetes communication"? Do you mean they differ in feeling the need to talk to a healthcare provider or different levels of stress?                       

Yes, regarding this, we have added a new reference on medical communication to illustrate that just different words may bring about differences in feelings.  

Reviewer 2 Report

1.This paper studied to explore the gender difference in diabetes distress among middle-aged and elderly diabetic patients. However, the number of 395 participants was not enough for this experiment.

2. It is lack of sufficient explanations in RESULTS and DISCUSSION.Try to explain your results in detail.

3. With minor revision as commented above, I recommend the publication in the Journal.

Author Response

  1. This paper studied to explore the gender difference in diabetes distress among middle-aged and elderly diabetic patients. However, the number of 395 participants was not enough for this experiment.

This article is a secondary analysis of the data collected during the development of TDDS. A sample size of 395 is an appropriate number of samples confirmed by the calculation of the number of samples during the development of the questionnaire. Therefore, although the number of samples may be slightly insufficient for the secondary analysis, we still considered it adequate for a valid analysis.

  1. It is lack of sufficient explanations in RESULTS and DISCUSSION. Try to explain your results in detail.

After taking into account the opinions of the four reviewers comprehensively, we decided to fine-tune the arrangement and presentation of the sentences to increase the clarity of the article rather than adding more sentences.

  1. With minor revision as commented above, I recommend the publication in the Journal.

Reviewer 3 Report

An interesting And very elaborate this article!
Is a practicat And useful research for phisician with different specialy:Internal Medicine, diabetology, Sportiv Medicine. The design And the method of the research are appropriate, with clear results  And good concordance with the aims. The References list list Is complete And relevant. The quality of english la guage is very good   , clear And concise. Interst of the readers Is one high VALUE because the interesting research theme And practicat results 

Author Response

An interesting and very elaborate this article!

Is a practicat and useful research for phisician with different specialy:Internal Medicine, diabetology, Sportiv Medicine. The design and the method of the research are appropriate, with clear results and good concordance with the aims. The References list is complete and relevant. The quality of English language is very good, clear and concise. Interest of the readers Is one high VALUE because the interesting research theme and practical results

Reviewer 4 Report

The results are good, but the manuscript is difficult to read. I recommend working on how the manuscript is written to make the presentation easier. The current manuscript has lots of very long compound sentences that make reading difficult.

Author Response

The results are good, but the manuscript is difficult to read. I recommend working on how the manuscript is written to make the presentation easier. The current manuscript has lots of very long compound sentences that make reading difficult.

Not sure what is meant by "four hospital with different levels"

We modified it to “four different-scaled hospitals” in Line 271 to correspond to the 4 different hospitals in Line 78.